# The Role of the Gut Microbiota in Colorectal Cancer Causation

**DOI:** 10.3390/ijms20215295

**Published:** 2019-10-24

**Authors:** Eiman A. Alhinai, Gemma E. Walton, Daniel M. Commane

**Affiliations:** 1Dietetics Department, Al Nahdha Hospital, Ministry of Health, P.O. Box 937, Ruwi, Muscat PC 112, Oman; Alhinaie78@gmail.com; 2Department of Food and Nutritional Sciences, University of Reading, Reading RG6 6UA, UK; 3Department of Applied and Health Sciences, University of Northumbria, Newcastle Upon Tyne NE1 8ST, UK; Daniel.commane@northumbria.ac.uk

**Keywords:** colorectal cancer, microbiota, *Fusobacteria*, *Bacteroides*, *Streptococcus Gallolyticus*, *Escherichia coli*, genotoxicity, gut

## Abstract

Here, we reviewed emerging evidence on the role of the microbial community in colorectal carcinogenesis. A healthy gut microbiota promotes intestinal homeostasis and can exert anti-cancer effects; however, this microbiota also produces a variety of metabolites that are genotoxic and which can negatively influence epithelial cell behaviour. Disturbances in the normal microbial balance, known as dysbiosis, are frequently observed in colorectal cancer (CRC) patients. Microbial species linked to CRC include certain strains of *Bacteroides fragilis*, *Escherichia coli, Streptococcus gallolyticus, Enterococcus faecalis* and *Fusobacterium nucleatum,* amongst others. Whether these microbes are merely passive dwellers exploiting the tumour environment, or rather, active protagonists in the carcinogenic process is the subject of much research. The incidence of chemically-induced tumours in mice models varies, depending upon the presence or absence of these microorganisms, thus strongly suggesting influences on disease causation. Putative mechanistic explanations differentially link these strains to DNA damage, inflammation, aberrant cell behaviour and immune suppression. In the future, modulating the composition and metabolic activity of this microbial community may have a role in prevention and therapy.

## 1. Introduction

The distinguishing characteristic of the colon is the relative abundance of the resident microbiota. Microbial communities present elsewhere in the GI tract are much smaller and show lower diversity. In parallel, the comparative incidence of colorectal to oesophageal and stomach cancers is approximately 30:1:2 [1,2]. Germ-free mice demonstrate the importance of host-microbe interactions to healthy host physiology; in these models, the colon architecture is visibly aberrant, there is an under-developed immune system [3], poor *wnt* signalling mediated differentiation of epithelial cells [4] and a functionally impaired epithelial barrier [5]. Further, the incidence of chemically induced tumours in mice models varies, depending upon the presence or absence of a functional microbiota [6].

Experimental intervention studies in non-germ-free animal models, with both probiotics and prebiotics, have been shown to suppress tumour development via diverse mechanisms. Several meta-analyses show that consuming a high-fibre diet reduces colorectal cancer (CRC) risk [7,8,9]. Fibre intake may be coupled to saccharolytic microbial activity in the gut and, in particular, the in situ synthesis of butyrate, with its well-studied anti-neoplastic activity. Thus, these strands of evidence indicate the importance of a healthy microbiota in cancer suppression. In contrast, we reviewed here the emerging evidence of the role of the microbial community in promoting colorectal carcinogenesis.

## 2. The Healthy Microbiota

In a healthy host, the colonic microbiome is typically dominated, at the phyla level, by Gram-negative *Bacteroidetes* and Gram-positive *Firmicutes,* with a smaller but sizable abundance of *Actinobacteria* and *Verrucomicrobia* [10,11]. The proportions of these phyla are not fixed, and different phyla, and indeed families, strains and species, compete to fulfil distinct ecological niches. Thus, under the influences of age, gender, genetics, diet and disease, there is considerable scope for inter-individual variation between phenotypically similar and healthy individuals [12]. Microbial diversity between individuals does not appear to critically influence central pathways in microbial metabolism. The fermentation of carbohydrates generally yields short-chain fatty acids, which can be used by the host, whilst proteolytic fermentation also yields phenols, cresols, ammonia and sulphides, commonly thought of as toxins. The production of specific secondary metabolites with pro and/or anti-carcinogenic activities, such as enterotoxins, cyclomodulins, B vitamins, urolithins, the estrogenic equol and mammalian lignans, may, however, be dependent on the abundance of certain strains, or functional groups, of bacteria. Equol, for example, is associated with a reduced risk of CRC [13], but is produced by fewer than 50% of the population and is dependent on colonisation with a handful of daidzein metabolising species [14].

## 3. The Microbiological Environment in Colorectal Cancer

Colorectal cancer has at least four recognised distinct common molecular subtypes [15]. Broadly speaking, cancers in the descending colon and rectum demonstrate high levels of chromosomal instability (CIN) and a strong up-regulation of *wnt* signalling [16]; in contrast, cancers of the ascending colon are rarer and are more likely to be of the microsatellite instability (MSI) subtype. Thus, the favoured anatomical distribution of these tumour sub-types hints at distinct aetiologies [17]. The right and left side of the colon have different embryological origins, but physiologically, these sections of the colon may be characterised as having distinct microbial activities. Saccharolytic fermentation dominates in the ascending colon, where the high fluid volume may also make the luminal contents quite dilute [18]. Microbial metabolites produced in the caecum, including short-chain fatty acids, may be reabsorbed, with water and electrolytes, in situ and through the transverse colon, such that the contents of the descending colon are more concentrated in biomass and potentially in toxic metabolites. In in vitro models, total microbial activity appears to decrease in the latter portions of the bowel and proteolysis becomes favoured [19,20]. Thus, distal and proximal colonocytes may be exposed to quite different microbial metabolites. To this point, these gradients in exposures have been poorly considered in relation to tumour subtype. Perhaps problematically, the aetiological/epidemiological studies continue to view CRC as a single disease, and therefore going forward, we will need to better consider tumour site and subtype in relation to diet and microbial exposures.

## 4. Microbial Metabolism in Carcinogenesis

Yachida et al. [21] compared the abundance of microbial metabolism genes present in stool between healthy controls and volunteers at different stages of the colorectal cancer process. In volunteers with preneoplastic polyps, they noted an increase in the abundance of genes involved in amino acid and sulphur metabolism and a relative decrease in the abundance of genes involved in methane metabolism. This is consistent with the long-standing assumption that a gut microbial environment favouring proteolytic over saccharolytic fermentation may increase CRC risk. With a western diet, somewhere between 6 and 18 g of protein per day is thought to reach the colon [22,23]. With decreasing availability of fermentable carbohydrate in the distal colon, there is a shift towards the production of proteolytic end products in the more cancer-prone left side [24]. In in vitro mixed culture models of gut fermentation, increasing protein concentrations in culture media leads to elevated production of phenolic compounds, amines, ammonia and hydrogen sulphide. These metabolites can be leveraged as nitrogen sources for bacterial cross-feeding, or they may be taken up by colonocytes and transported into the bloodstream [25]; however, their accumulation in the colonic lumen is associated with increased epithelial cell toxicity [26,27].

The amino acid composition of the protein substrate influences the overall composition of this potentially genotoxic fermentation supernatant. For example, methionine and cysteine may be used as a substrate by the sulphate-reducing bacteria (SRB) (i.e., *Desulfovibrio*, *Desulfotomaculum, Desulfobacter, Desulfobulbus* [28]), leading to the generation of H_2_S [28,29]. Hydrogen sulphide inhibits butyric acid oxidation [30,31,32], it increases cell proliferation in vitro [33] and is shown to be genotoxic [34]. In in vitro batch-culture fermentation with faecal inoculate, the rate of H_2_S production differs according to whether albumin or casein is used as a substrate [35]. In human observational studies, the sulphate reducing bacteria may be associated with inflammatory bowel disease [36] and are putatively implicated in its pathogenesis through the ability of H_2_S to compromise barrier function [29,37]. In both animal and human dietary intervention study, diets high in protein increase the recovery of sulphide in faeces [38], and Yachida et al. [21] observed an increase in the abundance of sulphate-reducing bacteria in stool samples from stage II and III cancer patients versus healthy controls.

Fermentation of aromatic amino acids leads to the production of phenols, indoles and 4-cresol. These are not well recovered in stool, but rather enter the hepatic circulation to be detoxified in the liver and eventually excreted in urine [39]. Studies have shown that with high protein intake, metabolites of 4-cresol and phenol appear in the urine [40]. Phenol and 4-cresol reach genotoxic concentrations in the in vitro gut fermentation models, which vary according to a protein source, and their concentration can be used to predict the genotoxicity of gut fermentation supernatants [41].

In contrast to the epidemiological data, the carcinogenicity of higher protein diets is consistently demonstrated, particularly in relation to colonic inflammation, in experimental animal models [42]. Higher protein dietary interventions in human volunteers lead to increased urinary excretion of markers of amino acid fermentation, but the appearance of these metabolites in urine does not necessarily correlate with increased faecal water genotoxicity [43,44]. Colonic fermentation and absorption are dynamic; therefore, faecal samples may be poorly representative of colonic exposures. Better biomarkers of cancer risk for human dietary intervention study are certainly needed to bridge the gap between the lack of associations between protein intake and cancer in human subjects versus the mechanistic and animal experimental evidence, implicating proteolytic gut fermentation metabolites in CRC.

## 5. The Colon Cancer Specific Microbiota

The tumour environment may be characterised by a disruption to the colonic stream, a depleted mucosal barrier, altered mucin secretion, inflammation, and changes in secretory IgA release that may facilitate aberrant biofilm formation. There may also be potential changes to the intestinal substrate in the form of blood, mucins, and host-derived lactate as a glycolytic metabolic by-product. Thus, specialists within the biota could well thrive as passengers in this ecological niche. The identification of tumour-specific microbes present in mucosal and/or faecal samples, and absent in healthy controls [45,46,47,48], or tumour tissue versus the adjacent healthy mucosa [49,50,51,52,53], prompts investigation into their precise role in disease aetiology. Enterotoxic strains of *Bacteroides fragilis*, PKS^+ve^ strains of *Escherichia Coli*, *Fusobacterium nucleatum, Enterococcus faecalis* and *Streptococcus gallolyticus* are candidate tumour-associated microbial species considered here for their roles in disease causality.

*Streptococcus gallolyticus*: Endocarditis and bacteraemia associated with *Streptococcus gallolyticus* (Sg) infection are associated with increased risk of colorectal neoplasia in observational studies [54,55]. Further, case-control studies show an increased risk of colorectal cancer associated with serological evidence of previous exposure to (Sg) antigen [56]. Faecal samples from volunteers with colorectal cancer are more likely to score positively for Sg, and tumour tissues show higher Sg counts than adjacent normal mucosa.

Experimentally, pre-exposure of cultured HCT116 cells to Sg prior to implantation resulted in the subsequent growth of a greater tumour mass in a mouse xenograft model. Further, oral gavage with Sg increased the tumour burden in AOM and DSS mouse models of tumorigenesis [57,58,59]. From a mechanistic perspective, Abdulamir et al. [60] observed a higher expression of Nf-KB and IL-8 mRNA in tumour tissues from individuals seropositive for Sg antibodies versus Sg negative patients, suggesting the Sg exposure induces a pro-inflammatory state, which may be a driver of cell turnover and is thus tumour promoting. However, Kumar et al. [57] observed increased cell proliferation in cultured colon cancer cell lines (HT29, HCT116 and LoVo) exposed to Sg and demonstrated that this was driven by an increase in nuclear β-catenin independent of inflammation. Further, there is evidence to suggest that the degree of tumorigenicity of individual sub-strains of Sg is dependent upon their ability to bind to intestinal epithelial cells [61]. Thus, adherent Sg may directly stimulate epithelial cell turnover through some, as yet not fully elucidated, cellular cross-talk.

*Enterococcus faecalis: E. faecalis* is closely related to *S. gallolyticus;* it has been associated with colitis [62] and colorectal cancer in observational studies [63,64]. Mechanistically, *E. faecalis* induces colitis in experimentally susceptible animal models [65,66]; however, experiments with cultured epithelial cells suggest that it may mediate the cancer process more directly through the production of genotoxic peroxide [67] and through its influences on cell cycle behaviour and the precipitation of polyploidy [68]. Recently, Lennard et al. [63] noted the greater expression of metastasis-associated genes in *E. faecalis* positive tumour tissue compared to non-EF colonised tumours, although follow on studies are yet to confirm this potential effect.

*Escherichia coli*: Raisch et al. [69] studied the abundance of *E. coli* by phylogenetic subgroup in mucosal biopsies from colorectal cancer versus mucosal samples from patients with diverticular disease as a control. They reported a much higher abundance of *E. coli* from the phylogenetic subgroup B2 in the cancer patient specimens (positively identified in 73.7% of cancer specimens versus 41.9% of controls). The phylogenetic B2 subgroup is home to enteropathogenic *E. coli* strains that are frequently associated with inflammatory bowel disease [62]; Fang et al. [70] observed specialist mucin degrading metabolic apparatus amongst the B2 *E. coli*, which may contribute to their colonisation and the pathogenesis of IBD. This group may also be characterised by the presence of genes encoding cyclomodulins, and genotoxins, such as colibactin. Cycle inhibiting factor (CIF) is a cyclomodulin capable of blocking mitosis independently of DNA damage, at least in vitro [71], and of inducing apoptosis in exposed epithelial cell lines [72]. In contrast, cytotoxic necrotising factor (CNF-1) exposure precipitates a reorganisation of the actin cytoskeleton and reversible cellular senescence that may be coupled to chromosomal irregularities and genomic instability in cultured colonocytes [73].

Colibactin is a poorly characterised genotoxic polyketide-peptide produced in the gut by polyketide synthase (PKS) positive *E. coli* [74]. These PKS positive bacteria have been identified in up to 20% of healthy volunteers. In animal models of carcinogenesis, exposure to PKS may induce DNA strand breaks [75] and tumour formation [76,77]. Transient infection of cultured epithelial cells with PKS positive *E. coli* induces chromosomal aberrations and increases the mutation frequency rate [75], in addition to influencing cell cycle behaviour [78].

*Bacteroides fragilis:* Perhaps up to 40% of the healthy population harbour intestinal strains of *B. fragilis* that are capable of producing a metalloproteinase enterotoxin (BFT). BFT^+^
*B. fragilis* colonisation has been associated with early neoplastic changes (adenoma and serrated polyps), but not necessarily in patients shown to have carcinoma [79,80], which might suggest a role in early carcinogenesis.

In mice, colonisation with BFT^+^
*B. fragilis* is capable of inducing Th-17-mediated colitis [81] and distal colorectal cancers in the APC^min^ mouse model in a manner dependent on IL17 mediated up-regulation of NF-κb, as demonstrated by Chung et al. [82] who observed repressed BFT-induced tumour formation in APC^min IL17/IL17^ mice. Further, Geis et al. [83] observed an accumulation of TReg cells in the APC^min^ mouse post-BFT colonisation, which may be the trigger for an IL17-mediated pro-carcinogenic inflammatory response.

Cultured epithelial cell models suggest alternative, or indeed complimentary, pro-carcinogenic mechanisms, the BFT toxin induces cleavage of E-Cadherin, leading to increased paracellular permeability and the activation of β-Catenin and a subsequent increase in cell proliferation [84], and it has been associated with polyamine metabolism-associated DNA damage when applied to cultured Ht29 and T80 cells [85]

It is also feasible that BFT^+^
*B. fragilis* facilitates localised microbial dysbiosis that indirectly facilitates carcinogenesis through the enabling of other pro-carcinogenic bacteria, both through its effects on the host immune apparatus [83] and potentially also through its effect on the gut barrier [84] and its recently reported influence on Muc2 synthesis and its well established role in mucin degradation [86]. Indeed, Dejea et al. [87] recently observed an abundance of biofilms co-colonised with BFT^+^
*B. fragilis* and PKS^+^
*E. coli* amongst FAP patients relative to healthy controls. They went on to demonstrate increased tumour lethality in co-colonised AOM-induced mice. Elsewhere, Drewes et al. [88] identified polymicrobial biofilms enriched with BFT+ *B. fragilis* and *Fusobacterium nucleatum* in 38 of 40 right-sided colorectal tumours but only 14 of 51 left-sided tumours. Tomkovich et al. [89] demonstrated that inoculation of APC min IL10-/- mice with polymicrobial biofilms isolated from tumour specimens induced tumours in the mice, importantly the BFT+ polymicrobial biofilms were more likely to induce inflammation in the mouse distal colon than BFT− samples. Interestingly, they also showed that a minority of healthy volunteers harboured colonic biofilms capable of inducing tumours in this model. This may have clinical implications for a faecal transplant, and further, given that we have limited direct proof of cancer causation, this group might warrant monitoring for future risk of colorectal cancer, and be leveraged for study in polyp recurrence studies.

*Fusobacterium nucleatum* is an oral symbiont, and occasional pathogen that has been identified in, and cultured from, intestinal tumours [50,90,91,92,93]. Yachida et al. [21] observed a progressive increase in the prevalence of *F. nucleatum* through the cancer stages beginning with highly dysplastic adenomas, whilst Amitay et al. [94] were unable to identify *F. nucleatum* in pre-neoplastic adenomas, suggesting that *F. nucleatum* is not a tumour initiator, but rather, an opportunistic coloniser during disease progression. Moreover, evidence suggests that *F. nucleatum* may be particularly associated with MSI and CIMP positive tumours [95]. Importantly, colonisation with FN has been associated with shorter survival in cancer patients in five out of 10 studies, as reviewed eloquently by Liu et al. [96], whilst in cultured cells, Rubinstein et al. [93] observed that *F. nucleatum* stimulates the proliferation of a panel of transformed colorectal cancer cell lines but not the non-cancerous (HEK293) cell line. In APC^min/+^ mice, inoculation with *F. nucleatum* induced the growth of significantly more tumours than in controls, whilst pre-treatment of tumour cells with *F. nucleatum* also induced significantly more tumours in a xenograft model [97]. Thus, in mice, *F. nucleatum* acts as a tumour promoter or enabler. A variety of potential tumour promoting mechanisms arebeing investigated. Tomkovich et al. [76,98] noted no increase in inflammation, in tumour susceptible APC^Min/+;Il10−/−^ and APC^Min/+^ germ-free animals colonised with *F. nucleatum,* suggesting that *F. nucleatum* does not promote tumourigenesis via inflammation-related pathways. Evidence is instead pointing to direct E-cadherin-mediated interactions with the epithelial cell *wnt* signalling pathway, thus promoting cell proliferation [93,97,99], and through suppression of immune surveillance [92,100,101].

## 6. Conclusions

Candidate tumour-associated microbes are still being identified. Here, we focused on those microbes best characterised in relation to CRC. These microbes are useful in that they demonstrate various roles for gut bacteria in different carcinogenic pathways across inflammation, immune suppression and through direct modulation of host cell behaviour. The proof of cancer causality for each of these microbes, in man, remains indirect; strong mechanistic studies are emerging to explain the observational associations with disease, as are studies showing causation in animal models (Table 1). Characterisation of banked stool samples from prospective cohort studies would strengthen the current evidence base further.

Questions also persist about the role of these tumour-associated microbes in different subtypes of CRC; *F. nucleatum,* for example, maybe more strongly associated with serrated adenomas [102,103], and BFT^+ve^ biofilms appear to dominate in right-sided tumours [88]. These observations may be a consequence of the wider gradient in microbial metabolic activity through the colon favouring certain bacteria in the ceacum; alternatively, it may be a reflection of higher toxic concentrations in the sigmoid colon favouring the chemically-induced mutations classically associated with the adenocarcinoma sequence. Future interventions aimed at modulating gut microbial metabolism in relation to CRC risk in man might also take into account differential drivers for specific tumour type and location. Important questions also remain about the role of specific microbiota at different stages of the cancer process.

In the future, targeted removal of early-stage carcinogenic members of the gut microbial community, perhaps with phage or other approaches, might be a desirable approach to reducing risk factors for cancer. Early-stage CRC-associated bacteria might also be useful as biomarkers of disease risk. On the other hand, transient colonisers, contributing later in the carcinogenic process, might be important drug targets, or where they are favourably adherent to tumour tissue they may have the potential for leverage as drug delivery vehicles.

## Figures and Tables

**Table 1 ijms-20-05295-t001:** Mechanistic evidence underpinning the putative role of tumour-associated gut bacteria in colorectal carcinogenesis. + suggestive evidence, ++ multiple strands of evidence.

	IBD Associated	Immuno-Suppressive	Pro-Inflammatory	CHROMOSOMAL instability	MSI Associated	CIMP Associated	DNA Damage Induction in Cultured Colonocytes	Proliferation Influence over Cultured Colonocytes	Metastasis Influencing
*Streptococcus gallolyticus*			++					++	
*Enterococcus faecalis*	++		++	++			++	++	+
CIF +ve *Escherichia coli*	++							++	
CNF +ve *Escherichia coli*				+				++	
Colibactin +ve *Eschericia coli*	++						++	++	
BFT+ve *Bacteroides fragilis*	++		++				++	++	
*Fusobacterium nucleatum*		++			++	++		++

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
