# Peer review of "The Role of the Gut Microbiota in Colorectal Cancer Causation"

_ijms, 2019, doi:10.3390/ijms20215295_

Round 1

Reviewer 1 Report

This is good review on importance of bacteria involvement in colorectal cancer. The review can be accepted with minor addition. Please extent review of clinical research in this field.

Author Response

We would like to thank the reviewer for their thoughtful review of our manuscript. 

The reviewer has suggested we consider extending the review of clinical research. 

Here we have been primarily attempting to capture the current state of understanding of the potential mechanisms of the microbiota in cancer causation.

In our opinion, the findings of observational studies in cancer patients are generally consistent; the same cancer associated species are reported again and again; thus we feel that presenting further observational studies would simply distract from the focus of this manuscript. 

Further given the explosion in the literature on the role of the gut microbiota in colorectal cancer, as such a more exhaustive review could be a very long document indeed and one not necessarily in the interests of the reader.

Reviewer 2 Report

In this manuscript Alhinai and co-workers reviewed the role of gut microbiota such as Bacteroides fragilis, Escherichia coli, Streptococcus gallolyticus, Enterococcus faecalisand Fusobacterium nucleatumin colorectal cancer. Both structure and content of manuscript were well organized and can be accepted for publication in this form. Only few typewriting errors were found:

Page 2 lines 48-49: Because of phylum of bacteria Bacteroidetes, Firmicutes etc. should be put in Italic style. Authors should put the cited reference number after term of et al.: for examples P2L82 Yachida et al., P3L106 Yachida et al., P4L157 Lennard et al. etc. P4lL52: dots (.) after E and S were missing. P4L166: instead of Coli, it should be E. coli. Names of journals in the list of cited references are inconsequent, for instant P7L286, P7L 292 etc. Revise them.

Author Response

We would like to thank the reviewer for their comments and for giving us the opportunity to improve our manuscript. 

To their points. 

1. Page 2 lines 48-49: Because of phylum of bacteria Bacteroidetes, Firmicutes etc. should be put in Italic style

Thank you, these have been changed accordingly.

2. Authors should put the cited reference number after term of et al.: for examples P2L82 Yachida et al., P3L106 Yachida et al., P4L157 Lennard et al. etc. P4lL52

we have performed a find and replace and amended these throughout. 

3. dots (.) after and S were missing. 

Again we have corrected these

4. P4L166: instead of Coli, it should be E. coli

We have corrected and performed a find and replace to ensure we did not miss any others

5.  Names of journals in the list of cited references are inconsequent, for instant P7L286, P7L 292 etc. Revise them

We have amended the reference format upon guidance from the editor. 

Finally, we have given the document a thorough editorial clean and hope that we have now removed any remaining typograhical errors.   

Kind regards 

Daniel Commane